# Diagnosis and Treatment of Inguinal Hernias after Surgical Treatment of Prostate Cancer, Current State of the Problem

**DOI:** 10.3390/jcm11185423

**Published:** 2022-09-15

**Authors:** Zhenghao Wu, Xinjian Zhang, Gilbert Charles Morgan, Bocen Li, Yuning Wang, Jiaming Wan, Yi Wang, Penghao Song, Yiyao Jin, Ruijie Zeng, Ming Wei, Chengyun Tang, Jin Zhang

**Affiliations:** 1Department of General Surgery, I.M. Sechenov First Moscow State Medical University, Ministry of Health of Russian Federation, 8/2 Trubetskaya Street, 119991 Moscow, Russia; 2School of Medicine, I.M. Sechenov First Moscow State Medical University, Ministry of Health of Russian Federation, 8/2 Trubetskaya Street, 119991 Moscow, Russia; 3Department of Pharmacology and Toxicology, University of Mississippi Medical Center, Jackson, MS 39216, USA

**Keywords:** TAPP, eTEP, inguinal hernia, prostate cancer

## Abstract

(1) Purpose: To compare and evaluate the immediate and long-term results of the use of various hernioplasties for the treatment of inguinal hernias after surgical treatment of prostate cancer; to determine the possibility of performing transabdominal preperitoneal (TAPP) hernioplasty and total extraperitoneal (eTEP) hernioplasty in patients with inguinal hernia during surgical treatment of prostate cancer. (2) Method: This study is a clinical analytical prospective study, without the use of randomization. The study included 220 patients with inguinal hernia, who were randomly divided into two groups (group A (*n* = 100) and group B (*n* = 120)). Patients in group A received eTEP, and those in group B received TAPP. The end points of the study were the results associated with the operation itself and the prognosis of the disease in the two groups. (3) Results: Group A: five patients had a scrotal hematoma, in 10 cases nosocomial pneumonia or infectious complications from the postoperative wound. The overall rate of early postoperative complications was 15%. In group B, the following postoperative complications were reported: one case of intestinal injury, six cases of acute urinary retention, eight cases of scrotal hematoma and 12 cases of nosocomial pneumonia or infectious complications from the postoperative wound were admitted. The overall incidence of early postoperative complications was 22.5%. There was no statistically significant difference in the incidence of postoperative complications between the two groups (χ^2^ (3) = 2.54, *p* > 0.05). (4) Conclusion: During the analysis of the obtained results, no statistically significant difference was found in the duration of hospitalization, the volume of blood loss, the severity of pain syndrome, postoperative complication incidence and recurrence incidence (*p* > 0.05); however, the comparison groups differed in the duration of the operation: the operation time in group A was much longer compared to group B (*p* < 0.05).

## 1. Introduction

Prostate cancer is a common neoplastic disease of men, and the frequency of its occurrence is increasing from year after year in parallel with the rate of aging of the population. It is a slow-growing, potentially lethal disease usually found in men over the age of 50. Although cases of the disease have been reported in all age groups, more than 80 percent of all prostate cancers occur in men over the age of 65. Prostate cancer risk factors include male gender, older age, positive family history, increased height, obesity, hypertension, lack of exercise, persistently elevated testosterone levels, Agent Orange exposure, and ethnicity [1,2]. The development and progression of prostate cancer is a complex process. The androgen-signaling pathway and its interaction with other pathways impact on cellular processes from growth, cell cycle, differentiation to growth arrest and apoptosis. Through adaptation and alteration, cells become tumorigenic [3].

The incidence of PCa diagnosis varies widely between different geographical areas. The highest incidence is recorded in Australia/New Zealand and North America (age-standardized rates (ASR) per 100,000 of 111.6 and 97.2, respectively), and in western and northern Europe (ASRs of 94.9 and 85, respectively), largely due to the use of prostate-specific antigen (PSA) testing and the aging population. The incidence is low in eastern and south-central Asia (ASRs of 10.5 and 4.5, respectively), but rising [4]. Rates in eastern and southern Europe were low but have also shown a steady increase [5]. Incidence and disease stage distribution patterns follow biological, genetic, and/or lifestyle factors. However, they are also influenced by (inter)national organizations’ recommendations on screening and diagnosis [6].

The widespread uses of clinical practice of prostate-specific antigen and transrectal prostate biopsy under the control of ultrasound significantly improved the indicators of early diagnosis of prostate cancer. At the same time, radical prostatectomy is still the method of choice for nonmetastatic prostate cancer, and postoperative survival after this procedure for one year reaches 96% [7]. The most common complications after radical prostatectomy are urinary incontinence and impotence. However, inguinal hernia (IH) is another recognized complication of radical prostatectomy, both laparoscopic- and robotic-assisted. The problems of urinary incontinence, erectile dysfunction and anastomosis strictures, as the main complications of the late postoperative period, are deservedly given much attention, while inguinal hernias after surgery for prostate cancer have remained in the shadows in the medical community for a long time [8]. Surgical site infection (SSI) is a frequent complication during urological surgery. Minimal invasive surgery (laparoscopic and robotic) is a good choice compared to traditional methods (open urological surgery) to reduce the risk of SSI [9]. Radical-assisted robotic prostatectomy is a commonly performed surgical procedure in the United States, which has low postoperative complication and recurrence rates [10].

Inguinal hernia repair is one of the most common surgical procedures in the world. About 20 million hernia repairs are performed each year. However, although laparoscopic inguinal herniorrhaphy began more than 28 years ago, most herniorrhaphy procedures are still performed using the open method [11]. Although the laparoscopic approach is widely recognized as an effective treatment for many diseases, and some laparoscopic surgical procedures have become the gold standard (e.g., cholecystectomy, appendectomy, and gastroesophageal joint surgery), the minimally invasive approach to inguinal hernia treatment remains highly controversial today. The main excuses were the high cost, the use of general anesthesia and the possible high incidence of major complications associated with laparoscopic surgery. Another silence point associated with the laparoscopic approach is the greater surgical complexity associated with the need to identify the “new” anatomy of the posterior groin wall, which is still unusual for the general surgeon. The choice of more laparoscopic methods (transabdominal preperitoneal (TAPP) versus complete extraperitoneal (TEP)) is also controversial [12]. Here, we introduce TAPP, focusing on tips and tricks for better results [13].

Arregui and co-authors proposed for the first time the preperitoneal placement of a mesh graft for its fixation to the suprapubic ligament and anterior abdominal wall using endoscopy [14]. Access to the preperitoneal space was carried out through the free abdominal cavity, and after fixing the mesh, the peritoneum was carefully sutured to reinforce the fascia to prevent the formation of adhesions in the abdominal cavity. This technique is called transabdominal preperitoneal hernioplasty (TAPP—transabdominal preperitoneal repair) and is currently widely used in the USA and western Europe [15]. TAPP is a safe and feasible surgical operation [16], It gives the surgeon a better view of inguinal anatomy and enables the surgeon to confirm the viability of the hernia content [16,17]. The complication of this method includes postoperative pain, mesh-related infection, mesh erosion to the bowel, etc. [18]. To minimize the risk of these complications, robot-assisted TAPP is a good choice to create intraperitoneal space in a minimally invasive way [18]. Despite a great outcome, the disadvantage of this procedure is its high degree of complexity: transperitoneal hernioplasty requires the surgeon to have excellent knowledge of anatomy and careful manipulation in the areas of vascular and nerve structures [19,20,21].

Complete extraperitoneal hernia repair (eTEP—totally extraperitoneal hernia repair) has been developed. The TEP was first performed in 1992 by Duluq to repair the inguinal hernia. The advantage of the operation is that it allows minimally invasive access without opening the peritoneum, which means that it can seal the hernia from the outside peritoneum to decrease the abdominal injury and decrease the risk of adhesion formation of intestines to prevent obstruction [22,23]. The extended totally extraperitoneal repair (eTEP) was developed in 2012 by Jorge Daes. The difference between eTEP and TEP is that there is a larger space to tackle large groin hernias [24,25].

## 2. General Information

From September 2017 to September 2021, 220 patients with inguinal hernia after prostate cancer treatment were selected. All patients were randomly divided into two groups: group A (*n* = 100) and group B (*n* = 120). Group A consisted of patients aged 18 to 80 years, the average age was 51.59 (±12.27) years. The types of inguinal hernias in the first group were distributed as follows: 92 cases with unilateral hernia, 8 cases with bilateral hernia, including 60 cases with oblique hernia, 40 cases with direct hernia. In group B, the age of the study participants ranged from 18 to 80 years, the average age was 53.07 (±15.71) years. Types of hernias: 90 cases with unilateral hernia, 30 cases with bilateral hernia, including 50 cases with oblique hernia and 70 cases with direct hernia.

## 3. Selection Criteria

Patients were selected for inclusion in the study according to following criteria.

### 3.1. Inclusion Criteria

(1)confirmed diagnosis of inguinal hernia(2)signed informed consent form(3)the possibility of performing surgical treatment

### 3.2. Exclusion Criteria

(1)the presence of dysfunction of the blood coagulation system(2)the presence of severe concomitant diseases(3)the presence of cognitive impairment or mental illness(4)the presence of severe generalized infections(5)lack of communication with the patient during the study

## 4. Methods

TAPP: in the inguinal region, 2 cm above and parallel to the inguinal fold, the skin and subcutaneous tissue are dissected to aponeurosis. The inguinal canal was then opened. A hernial sac located medially from the elements of the spermatic cord was isolated. The contents are an unchanged large omentum, the latter is immersed in the abdominal cavity, the transverse fascia above it is sutured with a continuous suture.

eTEP: after anesthesia, an incision was made above the navel transrectally on the left. The posterior leaf of the rectus abdominis muscle is separated, the preperitoneal space is open. A balloon is then introduced; the pre-bubble space is open. The supply of carbon dioxide has been adjusted under the control of IAP of 12 mm Hg. Optics No. 7 has been introduced. The hernial sac has been transferred to the preperitoneal space. The sac is located laterally. The pubic bone and inguinal ligament are completely isolated. An ultrapro 12 cm × 15 cm mesh is laid on the prepared platform, fixed by the hernia stapler. The preperitoneal space is drained; wounds are sutured, applying a bandage.

All patients were hospitalized for planned surgical treatment. At the prehospital stage, each of them underwent a comprehensive examination, including: a general blood test, a general urinalysis, a biochemical blood test (total protein, creatinine, urea, bilirubin, glucose, alanine and aspartic transaminases, C-reactive protein), a coagulation test (APTT, INR, prothrombin time, antithrombin III, fibrinogen), a blood test for HIV, TP, hepatovirus B and C. Of the instrumental studies, ECG, chest X-ray and abdominal ultrasound were mandatory. In the presence of indications (for example, an ulcerative history), esophagogastroduodenoscopy was performed. All patients over 40 years of age at the outpatient stage of the examination were consulted by a therapist to identify contraindications for planned surgical treatment under endotracheal anesthesia.

On the next day after surgery, all patients underwent an ultrasound examination to identify possible hematomas and seromas in the implant area, as well as mesh migration. Seromas and hematomas up to 20 mL in volume and spread out over the plane of the mesh were treated conservatively, and larger ones were punctured under ultrasound guidance. Control ultrasound was performed the next day after their removal. Two weeks after the operation, all patients of the prospective groups were examined on an outpatient basis in the clinical diagnostic center. The examination included a consultation with a surgeon and an ultrasound examination (detection of seromas, implant displacement).

The severity of pain in the early postoperative period was assessed using a visual analog pain scale (VAS). The patient was asked to answer the question about the severity of postoperative pain on a 10-point scale, where 0 is no pain, 10 is unbearable pain as shown in Figure 1.

To assess the long-term results of surgical treatment, questionnaires with a list of questions to which it was planned to receive answers were sent to all patients by mail (see the PATIENT QUESTIONNAIRE).
*QUESTIONNAIRE of a patient who underwent endovideosurgical hernioplasty for inguinal hernia**Dear patient! You underwent high-tech surgery for an inguinal hernia. In order to improve the efficiency of the surgical service of our medical institution, we ask you to answer a few questions. Send your answer by mail in the envelope attached to this letter. Thank you.**1. Surname, name, patronymic ____________________________________________**2. Gender: male female (circle as appropriate)**3. Age________**4. Date of operation________________**5. Type of hernioplasty____________________________ (not filled in by the patient)**6. Implant fixation method_________________________ (not filled in by the patient)**7. Duration of temporary disability after surgery (for employees), days: (circle as appropriate)**Up to 7**8–14**15–21**over 21**8. Time to return to physical activity after surgery: (circle as appropriate)**1–2 weeks**3–4 weeks**5–6 weeks**more than 6 weeks**9. Do you have any pain in the area of the operation at the moment? (circle as appropriate)**YES**NO**10. If YES, then rate their intensity on a 10-point system, where 0-no pain, 10-unbearable pain: (circle as appropriate)**0**1**2**3**4**5**6**7**8**9**10**10.1. Have you taken any pain medication for this pain? (circle as appropriate)**Daily (2–3 times a day)**Daily (1 time a day)**Periodically**Rarely**Very rarely**11. Are you satisfied with the result of the operation? (circle as appropriate)**YES (Completely)**YES (Partially)**NO**12. Did the hernia appear again in the same place? (circle as appropriate)**YES**NO*

The duration of surgical interventions was estimated in accordance with the protocols of operations and was expressed in minutes.

The duration of inpatient treatment of patients was expressed in days.

All data were entered into the patient’s individual card and subjected to statistical processing.

## 5. Indicators of Observation and Operation

To analyze the results of the study, the following end points were used: (1) the duration of the operation, (2) the length of stay in the hospital, (3) the volume of blood loss, (4) the severity of pain in the postoperative period, (5) measured on a visual analogue scale.

In addition to the main end points, the incidence of complications was analyzed, such as intraoperative intestinal trauma, acute urinary retention in the postoperative period, scrotal hematoma, infectious complications of postoperative wounds, and nosocomial pneumonia.

The recurrence rate was also assessed by monitoring patients for 6 months after discharge from the hospital.

## 6. Statistical Methods

SPSS19.0 software was used for data processing, measurement data were expressed as (x ± s) using the *t*-test; scoring data were expressed in % using the χ^2^-test; at *p* < 0.05, the difference was considered statistically significant (Table 1, Table 2, Table 3, Table 4, Table 5, Table 6 and Table 7).

## 7. Results

### 7.1. Comparison of Two Groups of Surgical Parameters

During the analysis of the obtained results, no statistically significant difference was found in the severity of the pain syndrome (*p* > 0.05).

### 7.2. Comparison of the Rate of Early Postoperative Complications between the Two Groups

In group A, there were five case of scrotal hematoma, 10 cases of nosocomial pneumonia or infectious complications of the postoperative wound. The overall rate of early postoperative complications was 15%.

In group B, the following postoperative complications were reported: one case of intestinal injury, 6 cases of acute urinary retention, 8 cases of scrotal hematoma, 12 cases of nosocomial pneumonia or infectious complications of the postoperative wound. The overall incidence of early postoperative complications was 22.5%.

The results of the chi-squared test of association revealed that there was no significant association in the incidence of postoperative complications between the eTEP group and the TAPP group (χ^2^(3) = 2.54, *p* > 0.05).

### 7.3. Comparison of Recurrence Rates between the Two Groups

The incidence of recurrence of the disease in group A was 0.00%, and in group B—2.5% (3/120). There was no statistically significant difference between the two groups for this indicator (χ^2^(1) = 2.535, *p* > 0.05).

## 8. Conclusions

The use of eTEP and TAPP hernioplasties for the treatment of inguinal hernias after radical prostate cancer surgery does not lead to an increase in the incidence of early postoperative complications in patients. For better performance of operations, it is necessary to conduct a preoperative check to determine the presence of inguinal hernia in patients with clinical manifestations or asymptomatic course. Patients with positive results should be informed in advance that a relapse may occur, and should also receive recommended preventive measures, to choose the appropriate methods of treatment. In the case of inguinal hernias in the postoperative period, it is necessary to carry out treatment appropriate to the physical condition of the patient and the severity of the inguinal hernia.

We recommend choosing the TAPP technique in the following cases: recurrent hernia, strangulated hernia; a history of operations on the organs of the lower floor of the abdominal cavity, for example, surgery on the prostate or bladder. This is because TAPP is a simple surgical method compared with eTEP, which does not require relatively extensive clinical experience.

Conditions for the use of eTEP: the hernial sac is small and easily completely removed; the presence of age-related changes and chronic pulmonary or heart failure. The advantage of eTEP is that its minimally invasive access can be achieved without opening the peritoneum. Although we found no statistically significant difference in the severity of the pain syndrome between these two methods, postoperative pain varied, especially after several months. In the case of chronic pulmonary or heart failure, patients who undergo eTEP will feel less pain because the surgical wound is less invasive. Moreover, eTEP is an expensive surgical method and requires extensive clinical experience.

TAPP and eTEP can be effectively and safely used in clinical practice for the treatment of inguinal hernias; however, each method has its own advantages and disadvantages. The choice of operation also depends on the specific situation and the condition of the patient. Further studies are needed to find out which method is more feasible in clinical practice.

## Figures and Tables

**Figure 1 jcm-11-05423-f001:**
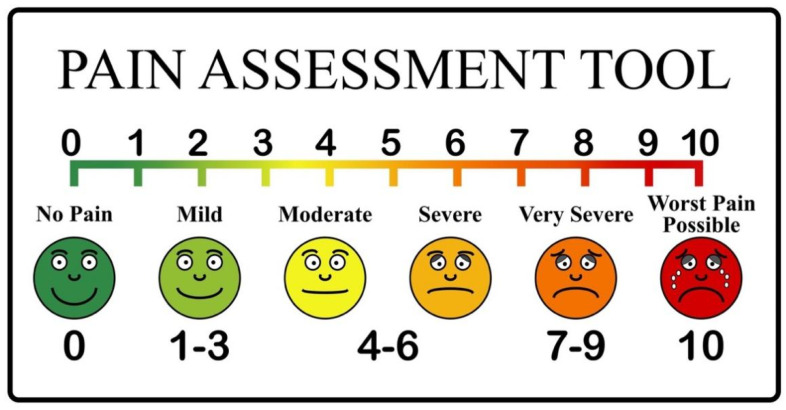
Pain Scale Chart: 1 to 10 Levels.

**Table 1 jcm-11-05423-t001:** Distribution of patients by age and method.

Method	Age (Year)									
	18–30		31–50		51–70		71–80		In Total	
	Number	%	Number	%	Number	%	Number	%	Number	%
TAPP	11	9.2	41	34.2	46	38.3	22	18.3	120	100
eTEP	4	4	41	41	50	50	5	5	100	100
Total	15	6.8	82	37.3	96	43.6	27	12.3	220	100

TAPP, transabdominal preperitoneal repair; eTEP, extended totally extraperitoneal hernia repair.

**Table 2 jcm-11-05423-t002:** Distribution by type of hernia.

Type of Hernia	TAPP (*n* = 120)		eTEP (*n* = 100)		In Total (*n* = 220)	
	Number	%	Number	%	Number	%
Unilateral hernia	90	75	92	92	182	82.7
Bilateral hernia	30	25	8	8	38	17.3
Total	120	100	100	100	220	100

TAPP, transabdominal preperitoneal repair; eTEP, extended totally extraperitoneal hernia repair.

**Table 3 jcm-11-05423-t003:** Comparison of the results of assessing the severity of pain according to VAS.

VAS Pain	TAPP	eTEP	*p*
2 h after surgery	4.58 ± 0.34	4.19 ± 0.28	0.271
The next day	4.05 ± 0.04	4.03 ± 0.17	0.364
At discharge	2.19 ± 0.22	2.03 ± 0.43	0.338
After a week	1.27 ± 0.15	1.09 ± 0.06	0.375
After months	0.59 ± 0.40	0.13 ± 0.74	0.298

VAS, visual analog pain scale; TAPP, transabdominal preperitoneal repair; eTEP, extended totally extraperitoneal repair.

**Table 4 jcm-11-05423-t004:** Comparative characteristics of postoperative complications or recurrence of hernia.

Postoperative Complication or Recurrence of Hernia	Age (Years)									
	18–30		31–50		51–70		71–80		In Total	
	TAPP	eTEP	TAPP	eTEP	TAPP	eTEP	TAPP	eTEP	TAPP	eTEP
Intestinal wounds	-	-	1	-	-	-	-	-	1	-
Hematoma	2	1	1	1	2	2	3	1	8	5
Other	3	2	3	1	5	3	7	4	18	10
Total	5	3	4	2	7	5	10	5	27	15

TAPP, transabdominal preperitoneal repair; eTEP, extended totally extraperitoneal hernia repair.

**Table 5 jcm-11-05423-t005:** Intraoperative and postoperative complications and unadjusted *p*-values.

Intraoperative and Postoperative Complications and Unadjusted *p*-Values			
Unadjusted Analysis	eTEP	TAPP	*p*
Intraoperative complications	80 (1.19%)	152 (1.40)	0.2763
Bleeding	53 (0.79%)	108 (0.99%)	0.1922
Injuries (total)	42 (0.63%)	77 (0.71%)	0.5705
Vascular	16 (0.24%)	34 (0.31%)	0.4662
bladder	3 (0.04%)	15 (0.14%)	0.0867
Bowel	4 (0.06%)	14 (0.13%)	0.2256

TAPP, transabdominal preperitoneal repair; eTEP, Extended totally extraperitoneal repair.

**Table 6 jcm-11-05423-t006:** Distribution of age comparison, PSA and stages of tumor in percentage (100%) in group A.

Number of Patients	Ages	Pre-Surgical PSA Levels	Stage of Tumor
20 (20%)	18–30	3 ng/mL	Stage I
30 (30%)	31–50	5 ng/mL	Stage I
40 (40%)	51–70	7 ng/mL	Stage II
10 (10%)	71–80	10 ng/mL	Stage II
Total = 100			

PSA, prostate-specific antigen.

**Table 7 jcm-11-05423-t007:** Distribution of age comparison, PSA and stages of tumor in percentage (100%) in group B.

Number of Patients	Ages	Pre-Surgical PSA Levels	Stage of Tumor
14 (12%)	18–30	2 ng/mL	Stage I
38 (32%)	31–50	5.5 ng/mL	Stage I
45 (37%)	51–70	8 ng/mL	Stage I
23 (19%)	71–80	9 ng/mL	Stage II
Total = 120			

PSA, prostate-specific antigen.

## Data Availability

Not applicable.

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
