# Peer review of "Diagnosis and Treatment of Inguinal Hernias after Surgical Treatment of Prostate Cancer, Current State of the Problem"

_jcm, 2022, doi:10.3390/jcm11185423_

Round 1

Reviewer 1 Report

General remarks:

- the sample size of 88 seems rather small. Did you do a power analysis?

- ref 4-14 are all cited in the same sentence. Please select max. 3. This is almost half of your total number of references!

- the paper is relatively short. In order to be able to replicate the experiment, more details need to provided. For example in section 5 (statistical methods). The 'Data Availability Statement' also needs to give more details (incl. a link to the data).

Introduction:

- "Prostate cancer is a common neoplastic disease of elderly men" -> This is an oversimplification. There are many younger men who get prostate cancer.

- More introductory text is needed: more background on prostate cancer, as well as similar studies that have been carried out.

- "some researchers have conducted an in-depth study of its frequency, factors, pathogenesis, as well as prevention and treatment measures." -> references to these studies?

Methods/results:

- There are paragraphs in the methods section that should be in the results section, and the other way around.

- Where is table 3?

Spelling/grammar remarks:

- Abstract: transabdominal preperitoneal hernioplasty (TAPP) and total extraperitoneal hernioplasty (TEP) -> transabdominal preperitoneal (TAPP) hernioplasty and total extraperitoneal (TEP) hernioplasty

- Introduction: "inguinal hernias ... has" -> "inguinal hernias ... have"

- Table 5: title seems to be incomplete?

Author Response

Dear Reviewers:

The attached document is our response, thank you.

Sincerely,

Jin Zhang

Reviewer 2 Report

In this manuscript, the authors analyze the feasibility and outcomes of transabdominal preperitoneal hernioplasty (TAPP) and extraperitoneal hernioplasty (TEP) in patients with an inguinal hernia after prostatectomy.

It is a good manuscript. Some revisions are needed.

  • Please improve epidemiology on PCa to better understand the wide diffusion of this malignancy (doi: 10.3390/app11041513)

  • With satisfaction, do you think an inguinal hernia could be reduced from the peritoneal or extraperitoneal cavity during a robot-assisted prostatectomy? (doi: 10.1089/end.2106.0225)

  • Is there a considerable risk of surgical site infection? May it be reduced from the mini-invasive surgery? (doi: 10.1515/med-2019-0081)

  • In the Conclusions section, how do you obtain the suggestions for TAPP and TEP from your work? Please clarify it

Author Response

(The authors gave the same response as above.)

Reviewer 3 Report

    The most common complications after radical prostatectomy are urinary incontinence and impotence. However, inguinal hernia (IH) is another recognized complication of radical prostatectomy, both laparoscopic and robotic assisted. In the present study, Wu and colleagues aim to compare two different approaches to the treatment of HI in terms of the incidence of postoperative complications: transabdominal preperitoneal repair (TAPP) and totally extraperitoneal hernia repair (TEP). The authors concluded that no differences were observed between the two techniques. In addition to the small number of patients, the lack of their characteristics (and a comparison between the two groups) make it difficult to understand the purpose of this work. For example, why are Tables 1 and 2 split? But above all, why did the authors try to group patients by gender???? Age, PSA level, Gleason Score, Class of risk... Some of these parameters could be listed and compared between group A and B. Authors concluded that TAP and TEPP should be considered for differents cluster of patients. Why? Please argue clearly this conclusion which is not evident from the results.  

Author Response

(The authors gave the same response as above.)

Round 2

Reviewer 1 Report

This version is a significant improvement over the previous version. I do have some minor issues that should be addressed.

Data availability statement: please refer to the exact data sources (link to the study page), not just TCGA and GEO. Or is the data not published yet, since it is a prospective study? See https://www.mdpi.com/journal/jcm/instructions#suppmaterials for instructions. This is important for the reproducibility of your research.

Informed consent statement: informed consent is one of the inclusion criteria, so please include your statement here. See https://www.mdpi.com/journal/jcm/instructions#ethics for instructions.

Line 235: I think end point 4 and 5 should be combined?

Some typos:

- line 84: "commom"-> "commonly"

- line 117: "ang" -> "and", "od" -> "of"

- line 119: "differnce"-> "difference", "ia" -> "is"

- line 294: "advantages" -> "advantage"

Reviewer 2 Report

The manuscript has been deeply revised. The introduction section goes through PCa epidemiology and its importance of wide diffusion. Hernia repair is often a challenging procedure after radical prostatectomy and the authors deeply analyze its treatment. In my opinion, it is a well-developed manuscript in this field.

Reviewer 3 Report

I appreciate the authors' efforts to improve the article.

However, the characteristics of the patients of the two groups are still missing. Please indicate and compare the age, the presurgical PSA, the stage of the tumors in the two groups, mainly because half of the patients look very young compared to the common age of prostate cancer. 
